# Global history, the emergence of chaos and inducing sustainability in networks of socio-ecological systems

Sabin Roman[1]*, Francesco Bertolotti[2]

**1** Centre for the Study of Existential Risk, University of Cambridge, Cambridge, United Kingdom, **2** LIUC - Carlo Cattaneo University, Castellanza, Lombardia, Italy

* sr911@cam.ac.uk

**Data Availability Statement:** All relevant data are within the paper and its Supporting information files.

**Funding:** This work is supported by the Grantham Foundation for the Protection of the Environment.

## Abstract

In this study, we propose a simplified model of a socio-environmental system that accounts for population, resources, and wealth, with a quadratic population contribution in the resource extraction term. Given its structure, an analytical treatment of attractors and bifurcations is possible. In particular, a Hopf bifurcation from a stable fixed point to a limit cycle emerges above a critical value of the extraction rate parameter. The stable fixed-point attractor can be interpreted as a sustainable regime, and a large-amplitude limit cycle as an unsustainable regime. The model is generalized to multiple interacting systems, with chaotic dynamics emerging for small non-uniformities in the interaction matrix. In contrast to systems where a specific parameter choice or high dimensionality is necessary for chaos to emerge, chaotic dynamics here appears as a generic feature of the system. In addition, we show that diffusion can stabilize networks of sustainable and unsustainable societies, and thus, interconnection could be a way of increasing resilience in global networked systems. Overall, the multi-systems model provides a timescale of predictability (300-1000 years) for societal dynamics comparable to results from other studies, while indicating that the emergent dynamics of networks of interacting societies over longer time spans is likely chaotic and hence unpredictable.

## Introduction

Socio-ecological systems are facing a multitude of problems related to the sustainable use of natural resources. Thus, developing tools for understanding and managing such complex systems is more crucial than ever to avoid ecosystem or social collapse [1]. Broadly, societal collapse can be defined as a "rapid and significant loss of an established level of socio-political complexity" [2, 3]. Theories on the mechanisms behind the phenomenon of collapse have not converged to a unifying framework, in part because they have historically been mostly qualitative in nature, and not aimed at a quantitative formulation that can help clarify problems and resolve tensions [4]. A number of models have been developed to address the long-term socio-environmental dynamics in different cases, such as Easter Island [5–10], the Maya civilization [11, 12], and other instances [13–18]. More theoretical approaches have developed models

The funders had no role in study design, data collection and analysis, decision to publish, or preparation of the manuscript.

**Competing interests:** The authors have declared that no competing interests exist.

focused on illustrating important concepts such as sunk-cost effects [19, 20], inequality [21], or sustainability more generally [22, 23].

This paper explores the dynamics of networks of socio-ecological systems by means of numerical simulation and investigates the conditions under which collapse occurs or is averted and overall sustainability can be achieved. More specifically, we propose a simplified model of a socio-ecological system that accounts for population, resources, and wealth, with a quadratic population contribution in the resource extraction term. The quadratic interaction is meant to represent the cooperative effects in the population (via self-interaction) working towards extracting resources and producing goods. While the model exemplifies the dynamics of a socio-ecological system, it is meant to capture general features over the long term and not reflect any specific real case study [24].

Given the structure of the model, an analytical treatment of attractors and bifurcations is possible, with a Hopf bifurcation from a stable fixed point to a limit cycle emerging above a critical value of the extraction rate parameter. The stable fixed point attractor can be interpreted as a sustainable regime, and the large amplitude limit cycle as an unsustainable regime. The model is then generalized to multiple interacting systems with chaotic dynamics emerging for small non-uniformities in the interaction matrix.

The chaotic behavior is described using usual numerical techniques, making use of the phase portrait, Poincaré map, and calculating the largest Lyapunov exponent. Certain choices of the interaction matrix lead to the synchronization of the systems in the network. In this case, the dimensionality of the problem reduces from $3N$ (where $N$ is the number of nodes in the network) to 3, and a chaotic attractor can be represented.

We introduce diffusion in the population variable, which can be seen as a type of migration under population pressure. We show that diffusion can stabilize networks of sustainable and unsustainable societies, and that interconnection could be a way to increase resilience in the global system. The work thus aims to contribute to "Global Systems Science," which previous work has advocated for [25].

In addition to the above mathematical considerations, we also present a novel interpretation of the dynamics in the Appendix in S1 Text As mentioned, systems reaching a steady state can be considered sustainable, but by analyzing the evolution of marginal costs and benefits, we can also conclude that they behave rationally (in economic parlance). On the other hand, systems where large amplitude limit cycles appear have marginal costs that exceed the marginal benefits, to the point where all net benefits are lost. Thus, these systems can be said to behave irrationally, and the framework presented provides a way to quantify long-term sunk-cost effects [19].

The paper proceeds as follows. In the next section, we briefly review the main methodological advances related to chaotic behavior in networked socio-ecological systems. Afterwards, we present the model for a single socio-environmental system and analyze it. We then explore the dynamics of multiple socio-ecological systems connected with different topologies. Finally, we draw the main conclusions of the paper.

## Background

Socio-ecological systems are typically complex and adaptive [26], where human and natural components interact [27]. They are known for exhibiting unpredictable behavior [28, 29], with sensitivity to initial conditions being one of the primary reasons for this behavior [30, 31]. This sensitivity is due to the presence of multiple feedback loops, both positive and negative, which can amplify small perturbations and lead to vastly different outcomes over time. For this reason, it is typically difficult to model trends pertaining to such systems [32], and it takes a long

time to assess the effect of actions on these systems [33]. A classic example of this phenomenon is the Lorenz attractor [34], where even small differences in the initial conditions of a simple mathematical model could lead to vastly different long-term behaviors. Conversely, no matter how much information is available on a system, if the dynamics are complex enough, it is commonly accepted that long-term predictions cannot be made. Unfortunately, even an apparently simple system of low dimensionality can be complex enough to generate this kind of behavior [35]. Moreover, socio-environmental systems are subject to unforeseeable events, such as natural disasters [36], emergent social inequality and technological breakthroughs [37, 38], which can introduce additional unpredictable features.

Ecosystem systems can be modeled as a dynamical system by means of a set of ordinary differential equations [39–41] that describe the flow of biomass throughout the class of entities that compose the system [42] or more generically, the exchange and consumption of stylized resources [43]. More precisely, this set of elements and connections can be modeled as a network, which behaviors can be chaotic [44].

Recently, it has been demonstrated that the emergence of chaos in a given networked system can be attributed to several key factors. First, an increase in dissipation within an ensemble characterized by a fixed coupling force and a set number of elements can initiate the onset of chaotic behavior [45, 46]. Additionally, modifications of network structure have the potential to engender chaos or even hyperchaos, exemplifying the sensitivity of these systems to alterations in their composition [25, 45–51].

It is well-known that chaotic behavior of network models could derive not only from the nodes' dynamics but also from the structure of the connections [52], especially in relation to links' orientation [53] and when the behavior is dissipative [48]. Also, the coupling strength between the elements plays a pivotal role in the manifestation of chaos, exhibiting a highly specific and nuanced influence on the overall dynamical behavior [46, 54]. Lastly, the heterogeneity of the components seems to play a role in the way chaotic dynamics arise [47, 55, 56]. Moreover, in ecological networks, chaotic behaviors are found for a wide set of dynamical processes, including percolation [57].

To the best of our knowledge, the emergence of chaos in socio-ecological networks mostly depends on three aspects. Firstly, the effect of coupling strength, which appears to be relevant both in ecological systems, where the unit of analysis is the biomass, and the relationships between nodes are competition or cooperation [45, 58], in social systems, where the link signifies stylized wealth relationships of various kinds [59–61], and in mixed socioecological systems [62]. Secondly, the effect of system efficiency, which counter-intuitively can have a negative effect on the stability and robustness of the system [63]. Similarly, the presence of chaos appears to be a necessary element for the functioning of resilient socioecological systems [64]. Thirdly, recent works used simulations to show how ecological-network stability is affected by their topology and functional dynamics, finding that size and the rate of connection affect the stability of a system [65], but that it cannot be solely determined by them [66].

Detecting chaos in a system is crucial for understanding its behavior. Ozer and Akin provide a comprehensive review of the main techniques employed for this purpose in their paper [67]. The most straightforward approach is time-series observation, which involves visually monitoring the system's state variables for signs of irregular or unpredictable behavior. Another useful method is phase portraits, which offer a two-dimensional representation of the phase space and can detect chaos by identifying distinct shapes. The Poincaré map simplifies complex behavior by sampling the phase portrait stroboscopically, replacing an $n_{th}$-order continuous-time system with an $(n-1)_{th}$-order map [53]. The presence of distinct points in the Poincaré map indicates chaos, making it ideal for stability analysis. Given the high

dimensionality of the system we are analyzing, we do not employ the Poincaré map directly but plot the distance to a given plane and observe the irregular points of intersection.

The power spectrum method analyzes frequency spectra of signals to identify chaos, which is characterized by wide-band signals that can be easily distinguished from periodic signals. Another common method to detect chaos is computing the Lyapunov exponents [68, 69], which measure the exponential attraction or separation in time of two adjacent orbits in the phase space with close initial conditions. When at least one Lyapunov exponent is positive, this is a strong indication that chaos is likely present.

## Model specification

We propose the following model:

$$
\begin{aligned}
\dot{x} &= \left( b \frac{z}{x + \epsilon} - dx \right) \left( 1 - e^{-z/\epsilon} \right) \\
\dot{y} &= ry \left( 1 - \frac{y}{K} \right) - \alpha x^2 y \\
\dot{z} &= \alpha x^2 y - cz
\end{aligned}
\tag{1}
$$

for a society with population $x$, natural resources $y$ and wealth (or capital) $z$. More generally, the model can be interpreted as follows: $x$ represents a bulk measure of capacity, such as extraction capacity, technological capability, state power, or working population. $y$ represents available resources, which are assumed to be regenerative, while $z$ represents returns. $x$ grows in proportion to returns per unit capacity and decays at a rate proportional to its size. $y$ grows logistically and decreases via extraction, in proportion to its current value times the square of the capacity (a measure of interactions within the social system). The extraction of resources leads to an increase in $z$. Table 1 provides details on the parameters of the model and their interpretations.

Note that the parameter $\epsilon > 0$ serves to regularize the behavior of the model when variables are close to the origin. The term $1 - e^{-z/\epsilon}$ in the first equation provides a regularization to this system when $z \approx 0$. It ensure that whenever $z \to 0$ it does so more smoothly and does not destabilize the numerical integration method. Without this term the networked dynamical systems are more stiff and more prone to numerical instabilities.

The capacity $x$ grows with a return-per-investment dependent net growth rate, and decays in proportion to its size. Resources $y$ recover logistically and are depleted in proportion to their current level and the capacity squared. The quadratic capacity term models interaction between units of extractive capacity (such as people or technology) and reflects the fact that

**Table 1. Description of parameters in model (1). The $\alpha$ and $\sigma$ parameters vary across the different scenarios.**

| Parameter symbol | Parameter name | Typical value(s) |
| --- | --- | --- |
| $b$ | Capacity growth rate | 2 |
| $d$ | Capacity decay rate | 0.05 |
| $r$ | Resources regeneration rate | 0.05 |
| $K$ | Maximum resource level | 100 |
| $\alpha$ | Extraction rate of resources | $2.5 \times 10^{-5}$ |
| $c$ | Returns decay rate | 0.05 |
| $\epsilon$ | Regularisation parameter | 0.01 |
| $\sigma$ | Diffusion constant | 0.1 |

multiple cooperating units are at play within a extraction-production-distribution chain. The extraction of resources contributes to the returns $z$, and they deplete in proportion to their size. We expect the capacity and returns grow concomitantly, so the depletion term reflects both higher consumption and natural decay over the lifespan of the goods. In Appendix A in S1 Text, we provide an economic interpretation of the model and the dynamics of collapse.

In a similar spirit to other models [7], the system (1) has a wealth (or capital) variable, that can be interpreted as the number of cultural artefacts of that society, such as buildings and food reserves. Furthermore, the model has the general structure of the Easter Island model [10] or HANDY model [21], and its dynamics echoes those models as well. However, the structure and analysis of the model is much simpler, which allows us to better highlight its economic interpretation. More importantly, the quadratic capacity in the extraction terms allows the introduction of interactions between multiple regions. This feature of the model allows for rich dynamical behaviour in higher dimensions. The systems of differential equations were solved using the 4th order Runge-Kutta method.

## Model analysis

The fixed points of the system (1) are the origin $O = (0, 0, 0)$, the point $N = (0, K, 0)$ and $E = (x_{eq}, y_{eq}, z_{eq})$, where:

$$
\begin{aligned}
x_{eq} &= \sqrt{\frac{r}{\alpha}\left(1 - \frac{y_{eq}}{K}\right)} \\
y_{eq} &= \frac{cd}{\alpha b} \\
z_{eq} &= \frac{d}{b}x_{eq}^2
\end{aligned}
\tag{2}
$$

The long-term behaviour depends on the $\alpha$ parameter. Let:

$$
\alpha_\star = \frac{cd}{Kb}
\tag{3}
$$

If $\alpha < \alpha_\star$ then the point $N$ is a stable fixed point but for higher $\alpha$ a trans-critical bifurcation occurs and $E$ becomes stable. At $\alpha = 2\alpha_\star$ the capacity (or population) is at the maximum sustainable level. The system shows a supercritical Hopf bifurcation for the extraction rate given by:

$$
\alpha_c = \frac{2cd + (c + 2d)^2}{4Kb}\left(1 + \sqrt{1 + \frac{8rcd(c + 2d)}{[2cd + (c + 2d)^2]^2}}\right)
\tag{4}
$$

Fig 1 shows the dynamics of the system (1) when most of the parameters are kept constant except for the extraction rate $\alpha$, which is varied. A limit cycle emerges for $\alpha > \alpha_c$, as highlighted by the analysis. In Fig 1(D), we can see that the amplitude of the oscillations reduces with increasing $\alpha$.

For the extraction rate in the interval $\alpha_\star < \alpha < \alpha_c$, the system (1) reaches a steady state, and we refer to systems in this regime as sustainable because the population, resources, and wealth do not collapse to zero values. On the other hand, for most $\alpha > \alpha_c$, the system (1) has oscillations that approach the origin, and we label these systems as unsustainable.

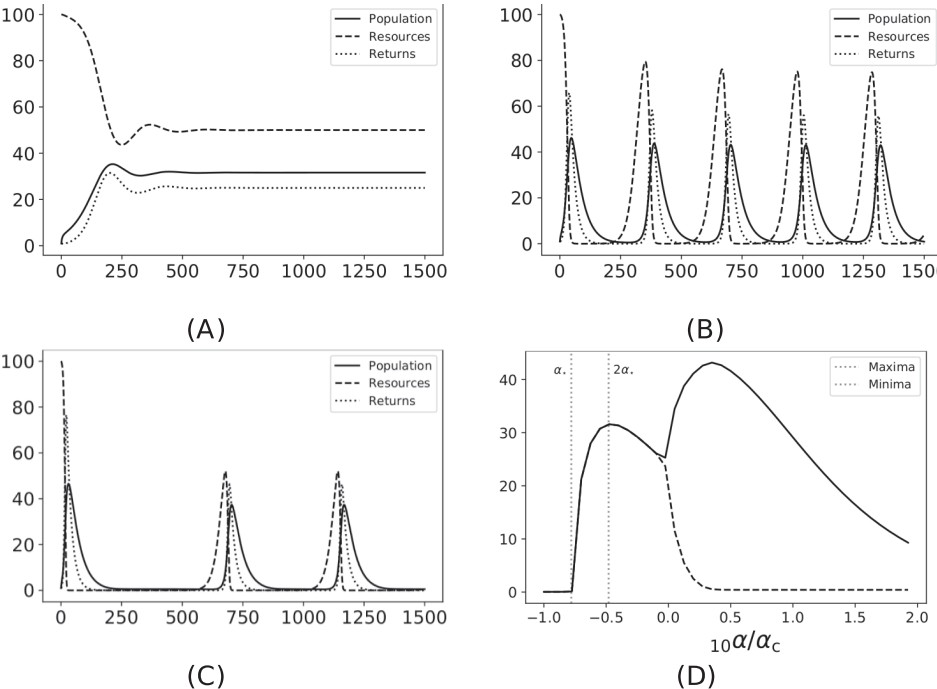

**Fig 1.** Trajectories converging to different attractors: (A) inner stable fixed point for $\alpha = 2\alpha_\star$, (B) limit cycle for $\alpha = 2\alpha_c$, (C) oscillations spread out over time and reduce in amplitude, $\alpha = 5\alpha_c$. (D) Bifurcation diagram showing minimum and maximum value of the population (or more generality the capacity) for increasing values of $\alpha$. The vertical lines (dotted) indicate the $\alpha_\star$ and $2\alpha_\star$ values. The initial conditions we took are $(x_0, y_0, z_0) = (1, 100, 1)$.

## Dynamics on networks

We generalize the system (1) to a network model in the following way:

$$
\begin{aligned}
\dot{x}_i &= \left( b\frac{z_i}{x_i + \epsilon} - dx_i \right)\left(1 - e^{-z_i/\epsilon}\right) - \sigma \sum_{j=1}^{N} x_j B_{ij} \\
\dot{y}_i &= ry_i\left(1 - \frac{y_i}{K}\right) - \alpha_i x_i y_i \sum_{j=1}^{N} x_j A_{ij} \\
\dot{z}_i &= \alpha_i x_i y_i \sum_{j=1}^{N} x_j A_{ij} - cz_i
\end{aligned}
\tag{5}
$$

The matrix $B$ is the Laplacian for the undirected network we are investigating (the node degrees are on the diagonal, −1 for each edge connection, otherwise zero), and in the case where $x_i$ represent populations, $B$ is meant to represent migration as diffusion. For generality, we consider the extraction network and migration networks to be separate. We use the matrix $A$ to represent the interaction between multiple societies with regards to extraction of resources.

## Emergence of chaos

We first consider the case where there is no diffusion, so $B = 0$, and $A$ is a matrix with entries equal to $1/N$, where $N$ is the number of subsystems. Let $1_N$ be a $N$-length column vector of 1s.

Then, we can write $A = 1_N 1_N^T / N$. In this case the dynamics of the system (5) is the same as for the system (1), namely all subsystems are synchronised and have the same bifurcations shown in Fig 1(D).

Nevertheless, if we take $A = 1_N 1_N^T / N + \delta I$ where $\delta$ is a small perturbation (we take $\delta = 10^{-2}/N$) and $\alpha > 3\alpha_c$, then we see the emergence of chaotic behaviour. All the $N$ subsystems still synchronise but are no longer attracted to the limit cycle in Fig 1(B) and 1(C). Instead, the trajectories follow the chaotic attractor in Fig 2(A).

In Fig 2(B) the distance to the plane $\sum_1^N x_i + y_i + z_i = N(x_{eq} + y_{eq} + z_{eq})$ is plotted and is a way to represent the Poincare map of the system (5). We see the distance has a periodic behaviour up to a point and then becomes aperiodic. Fig 2(C) shows the bifurcation diagram for the $x_1$ variable where we see the emergence of multiple extrema when $\alpha > 3\alpha_c$, which is indicative of chaos. In Fig 2(D) the spectra of the $x_1$ time series is shown, contrasting the case of periodic behaviour (with well-separated peaks) and that of chaos (with a high density of peaks). Fig 2 (E) shows two trajectories starting from initial conditions differing in all components by $10^{-9}$. In Fig 2(F), we plot the distance between these trajectories and notice a exponentially growing divergence, indicating a positive Lyapunov exponent, as expected for a chaotic system.

The largest Lyapunov exponent for the parametrization we employed is estimated to be $3.4(2) \times 10^{-3}$. The inverse of the exponent gives a timescale of 290±30 years, which is compatible with the average lifespan of empires of 220 years [70]. Different parameter choices (of $\alpha$ and $A$) give rise to other timescales in the range of 300 to 1000 years.

This suggests that societal dynamics has a time horizon of predictability within these bounds, and these theoretical estimates also agree with the duration of different complex societies and empires [71]. The emergence of unstable behaviour on different networks is well documented in the scientific literature [51, 55, 72] and our results fall in this domain.

## Inducing sustainability in a network

Next, we consider the case where $A = I_N$, so the subsystems are independent in their extraction efforts, and we have different topologies for migration (via diffusion) to take place. In Fig 3, we see the different networks considered on the left side and the ratio of the maximum and minimum total population on the right side.

The diffusion parameter $\sigma$ is varied in the interval [0, 0.3] and the population ratio is plotted throughout. A sustainable society (white node) always has a fixed extraction rate of $\alpha = 2\alpha_\star$ and for unsustainable societies (black nodes), we consider different extraction rates higher than $\alpha_c$ by factors of 1.5, 2 and 2.5 respectively.

For the chain network in Fig 3(A), we see that if $\alpha/\alpha_c = 1.5$ for the black nodes, then there is a range for higher values of $\sigma$ with a max/min population ratio of 1 (see Fig 3(B), solid line). This indicates that there is no limit cycle and that all populations in the network reach a fixed point. If $\alpha/\alpha_c$ is increased, oscillations appear, with the ratio between the maximum and minimum total population beyond 2 or 3. In Fig 3(C), we have a ring topology with only one sustainable society. As shown in Fig 3(D), a similar situation arises, where for $\alpha/\alpha_c = 1.5$ there is a range where the system reaches a fixed point, whereas for higher extraction rates, large oscillations appear.

For the star topology in Fig 3(E), despite all the neighbors of the white node being unsustainable, there still exists a range of $\sigma$ for $\alpha/\alpha_c = 1.5$ in Fig 3(F), where a fixed point appears for the entire network (i.e., global sustainability exists for the whole network). Higher extraction rates for the black nodes again lead to large oscillations.

In Fig 3(g), we investigate whether coupling sustainable societies leads to stronger sustainable outcomes. As shown in Fig 3(h), the range where a global sustainable outcome is reached

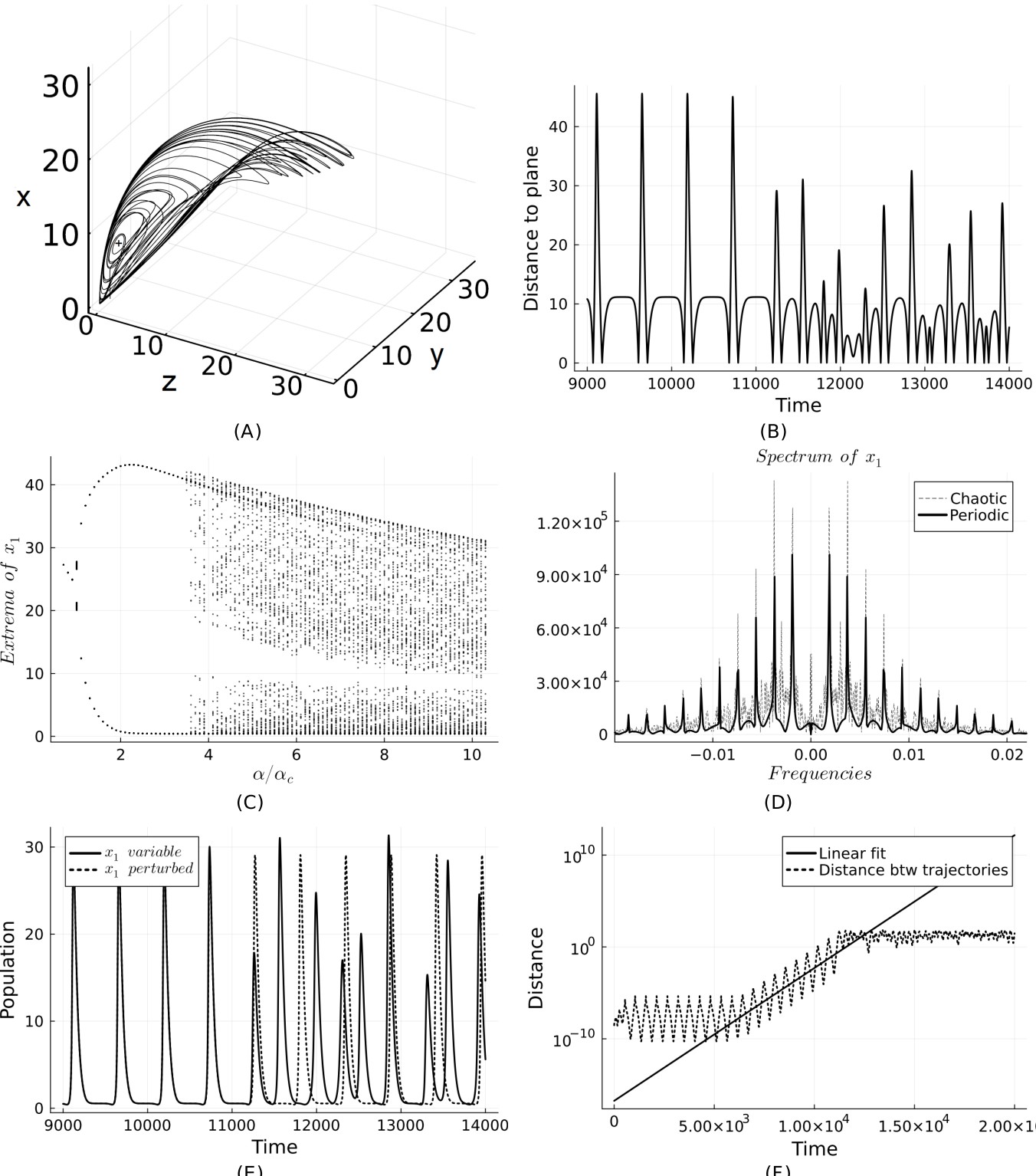

**Fig 2.** (A) Chaotic attractor for the synchronized system (5) with $N = 3$, $A = 1_N 1_N^T / N + \delta I$ and $\alpha = 10\alpha_c$. (B) Distance to the plane $\sum_1^N x_i + y_i + z_i = $ const., meant to represent intersections giving the Poincaré map. (C) Bifurcation diagram showing the extrema of the $x_1$ variable where $\alpha$ is varied. (D) Spectrum of the time series of $x_1$ showing the difference between the frequencies prevalent in two regimes: periodic (with well-separated peaks) versus chaotic (high density of peaks). (E) Divergent trajectories for initial conditions differing by $10^{-9}$ (in all components). (F) Estimate of the largest Lyapunov exponent, giving $3.4(2) \times 10^{-3}$.

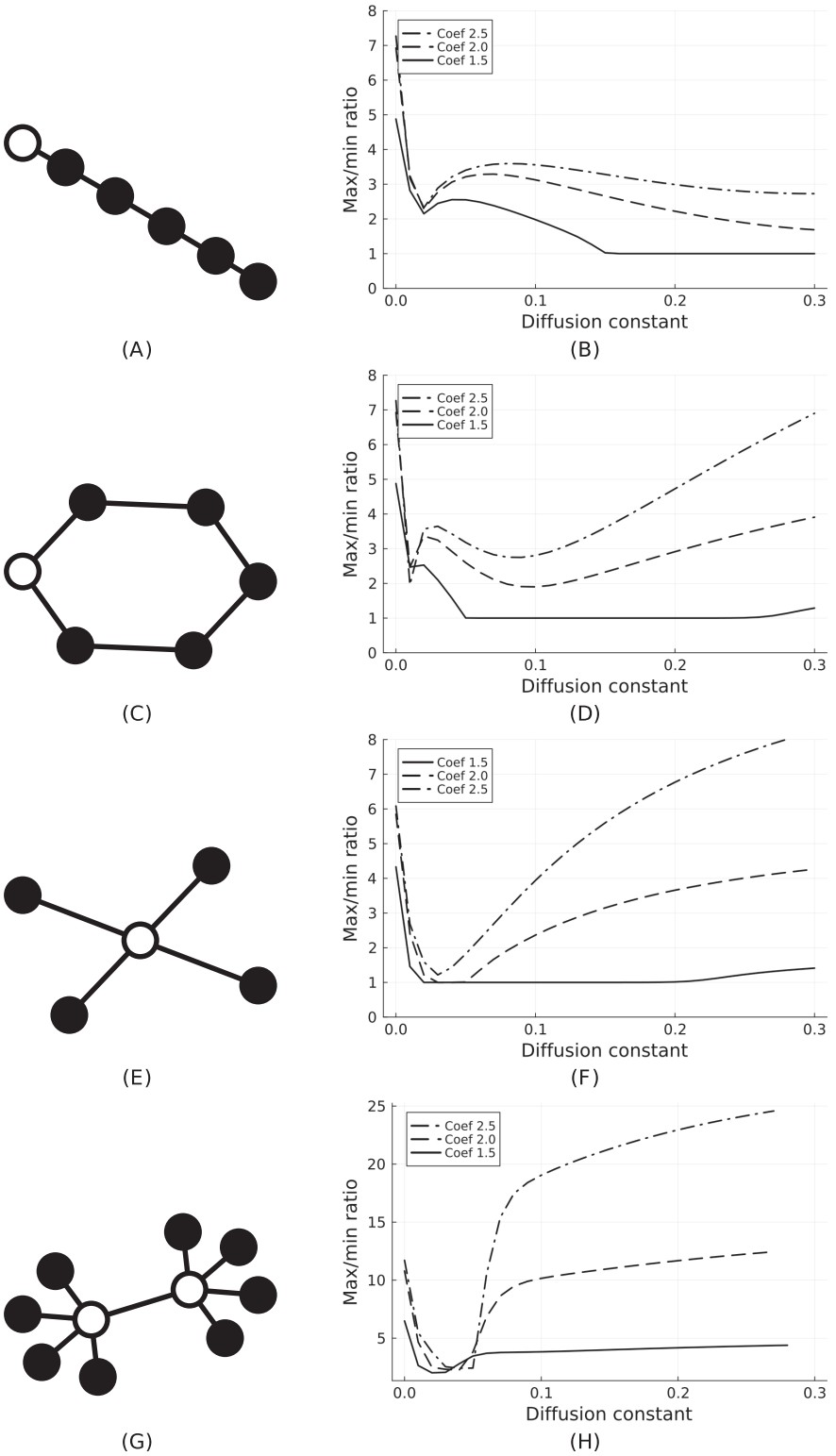

**Fig 3.** (Left column) Different topologies of networked socio-environmental systems. The white nodes are systems with the default parameters from Table 1 and extraction rate $2\alpha_*$, while the black nodes have extraction rate $Coef \times a_c$ (see the legends in the right column). (Right column) The ratio of the maximum and minimum total population on the attractor for the networked systems on the left. The diffusion constant $\sigma$ is varied and the black nodes have extraction rates $\alpha/\alpha_c = 1.5$ (solid line), 2 (dashed line) or 2.5 (dashed-dot line). If the ratio is 1 we get a fixed point.

(fixed point) is reduced compared to Fig 3(E) and 3(F), and limit cycles with large amplitudes arise over most of the parameter space (for $\sigma$ and $\alpha/\alpha_c$).

The findings suggest that in a random graph (of average degree four), a fraction of 20% of sustainable systems can lead to a sustainable outcome for the entire network (provided that sustainable systems are not directly connected to each other). Overall, depending on the topology, a relatively small fraction of sustainable societies can either reduce oscillations in the overall system or even induce a global sustainable regime.

## Conclusion

We have proposed a three-dimensional model of socio-environmental dynamics that accounts for population, resources, and wealth (or capital). The model is inspired by prior work [10, 21] but differs in important ways; specifically, it has an overall simpler structure along with a quadratic interaction term that captures the cooperative efforts involved in resource extraction and production of goods. An analysis of its dynamics shows that societies with an extraction rate below a critical threshold will reach a steady state. If the extraction rate is above a critical value, then supercritical Hopf bifurcation leads to the appearance of a limit cycle. The oscillations are typically large enough to approach the origin; therefore, all variables are periodically at almost zero values (i.e., the system collapses). Hence, systems with a low extraction rate of natural resources that reach a steady state are called sustainable, whereas systems with a high extraction rate that oscillate with a large amplitude are considered unsustainable.

We also provide an economic interpretation of collapse in the Appendix in S1 Text and show that for sustainable systems, the marginal benefits and costs of growth equalize, leading to a rational, optimized outcome. However, for unsustainable societies, it is not the marginal benefits and costs that equalize but the overall cumulative amounts. Hence, according to this specific economic rationale, unsustainable societies are irrational. This sensitivity to the cumulative benefits and costs indicates a way to quantify long-term sunk-cost effects that are drivers of collapse [19].

Two extensions of the model to networks are considered: one involving an interaction matrix in the extraction (and production) terms and another involving diffusion in the population variable (akin to migration). If the interaction matrix is uniform, the dynamics matches the behavior of the single system. However, even small perturbations in the interaction matrix are sufficient to generate chaotic dynamics (provided that the extraction rate is above a certain threshold). If only the diagonal terms are altered in the same way, then the systems in the network synchronize, and the overall dimensionality of the system is reduced to three; in this case, we can visualize the emerging chaotic attractor. The inverse of the Lyapunov exponent provides timescales consistent with the historical dynamics of empires [70, 71].

For the diffusion case, we find that for a variety of network topologies (chain, ring, and star), a small fraction of sustainable systems is sufficient to drive the entire network to a steady state. Similar results have previously been found for Easter Island [10] in the case of two coupled societies but in a more restricted parameter range. The results in the present paper generalize and strengthen the previous findings to multiple societies and larger parts of parameter space.

Linking sustainable societies does not increase network effects that favour steady states, but seems to diminish them. Overall, in a random graph (of average degree four), a fraction of 20% of sustainable systems (not directly connected to each other) can lead to a sustainable outcome for the entire network. Other topologies, community structures, and their impact on the sustainability of the overall system will be the subject of future work. Thus, our work aims to contribute to a "Global Systems Science" [25].

## Supporting information

**S1 Text. Appendix A.** In the appendix we propose an economic interpretation for model (1). Sunk costs effects [19] are quantified and compared with the definition of rationality in economics as utility maximization.
(PDF)

## Author Contributions

**Conceptualization:** Sabin Roman.

**Formal analysis:** Sabin Roman.

**Investigation:** Sabin Roman.

**Methodology:** Sabin Roman, Francesco Bertolotti.

**Project administration:** Sabin Roman.

**Resources:** Francesco Bertolotti.

**Software:** Sabin Roman, Francesco Bertolotti.

**Validation:** Francesco Bertolotti.

**Writing – original draft:** Sabin Roman, Francesco Bertolotti.

**Writing – review & editing:** Sabin Roman, Francesco Bertolotti.

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
