## [Decision Letter · Decision Letter 0]

30 Jun 2023

PONE-D-23-11323Global history, the emergence of chaos and inducing sustainability in networks of socio-ecological systemsPLOS ONE

Dear Dr. Roman,

Thank you for submitting your manuscript to PLOS ONE. After careful consideration, we feel that it has merit but does not fully meet PLOS ONE’s publication criteria as it currently stands. Therefore, we invite you to submit a revised version of the manuscript that addresses the points raised during the review process.

We look forward to receiving your revised manuscript.

Kind regards,

Pilwon Kim

Academic Editor

PLOS ONE

Journal Requirements:

“This work is supported by the Grantham Foundation for the Protection of the 266 Environment.”

Reviewers' comments:

Reviewer's Responses to Questions

**Comments to the Author**

1. Is the manuscript technically sound, and do the data support the conclusions?

Reviewer #1: Yes

Reviewer #2: Yes

2. Has the statistical analysis been performed appropriately and rigorously? 

Reviewer #1: No

Reviewer #2: Yes

3. Have the authors made all data underlying the findings in their manuscript fully available?

Reviewer #1: Yes

Reviewer #2: Yes

4. Is the manuscript presented in an intelligible fashion and written in standard English?

Reviewer #1: Yes

Reviewer #2: Yes

5. Review Comments to the Author

Reviewer #1: The manuscript presents a new dynamical system to model socio-environmental systems with three state variables representing population, resources and wealth. The system is based on a previous model (Easter Island model), but is simpler. A bifurcation diagram based on the extraction rate of resources leading from steady state to limit cycle via Hopf bifurcation and to chaos is presented; a network of coupled oscillators is defined and studied based on a diffusion parameter and the topology. The results delimit parameter ranges where the system becomes sustainable, i.e., the population reaches a nonzero steady state. The oscillatory behavior is to be avoided, since it shows minimum values close to zero, where the system collapses. Overall, the paper is cleary written and the results seem to be correct. I have a few corrections that I would like to see addressed before the paper can be accepted.

1) It is not clear to me if the authors consider the new system as more realistic or not in comparison to Easter Island. Please, comment on that.

2) The terms in the model equation (Equation 1) are well explained, but I did not understand the meaning of the term -e^(-z/eps) in the equation for dx/dt.

3) In the paragraph right after Eq. (1), "The decrease in resources leads to an increase in returns z". Since alpha>0, based on the equation for dz/dt I think it should be "The increase in resources leads to an increase in returns z".

4) In Fig. 1(d), it would help the reading to indicate the position of alpha* and 2alpha* in the bifurcation diagram. It becomes clear in the text, but a visual mark would make it easier for the reader.

5) In the caption of Fig. 1, it should be alpha=2alpha*, instead of alpha=2a*.

6) In table 1, how were the typical values chosen?

7) On page 6, A is defined as a matrix with entries equal to 1/N, but in the mathematical definition, A=1_N 1_N^T/N. I am confused with the notation. Assuming that 1_N is an NxN matrix with ones in all positions, then 1_N * 1_N^T = 5_N, so A=1 in all positions. For me, it should be A=1_N/N.

8) On page 6, second paragraph, "We see the distance has a periodic behavior up to a point and then becomes chaotic". Since chaos has not been demonstrated up to this point in the text, I suggest "We see that the distance has a periodic behavior up to a point and then becomes aperiodic".

9) Based on Fig. 2d, the computation of the maximum Lyapunov exponent seems to be based on a single pair of initial conditions, and a short time series. The Lyapunov exponent is an asymptotic result of ergodic systems, so the time series should be long. Since the nearby trajectories will inevitably diverge from the close vicinity of each other in chaotic systems, people usually periodically reorthogonalize the trajectories and rescale their distances in order to keep computing the Lyapunov exponent for a long time (see J. C. Sprott https://sprott.physics.wisc.edu/chaos/lyapexp.htm). Alternatively, the average result from an ensamble of initial conditions should be computed.

10) Since matrix B determines the network topologies shown in Fig. 3, I would like to see the actual matrices employed for each network. That makes it easier to reproduce the results.

11) The bibliographic references need to be revised. Some errors are pointed below:

Reference 14 - Publisher's name is omitted. I think it is Oxford University Press.

Reference 24 - Journal information is omitted. It is Nature 497, 51-59.

Reference 28 - Publisher's name is omitted.

Reference 33 - There are invalid characters in the doi code.

Reference 38 - page information is strange.

Reference 59 - volume and page information are missing.

Reference 65 - Publisher's name is omitted.

Reference 67 - Publisher's name is omitted.

Reference 71 - Volume information is missing.

Reviewer #2: In this work, the authors have proposed a three-dimensional model of socio-environmental dynamics that accounts for population, resources and wealth. The topic is hot. The paper is well structured and written. The obtained results are new and very interesting. So, this work merit to be published.

6. PLOS authors have the option to publish the peer review history of their article (what does this mean?). If published, this will include your full peer review and any attached files.

Reviewer #1: No

Reviewer #2: No

---

## [Editor Report · Decision Letter 1]

12 Oct 2023

Global history, the emergence of chaos and inducing sustainability in networks of socio-ecological systems

PONE-D-23-11323R1

Dear Dr. Roman,

We’re pleased to inform you that your manuscript has been judged scientifically suitable for publication and will be formally accepted for publication once it meets all outstanding technical requirements.

Kind regards,

Pilwon Kim

Academic Editor

PLOS ONE
---

## [Editor Report · Acceptance letter]

4 Nov 2023

PONE-D-23-11323R1 

Global history, the emergence of chaos and inducing sustainability in networks of socio-ecological systems 

Dear Dr. Roman:

I'm pleased to inform you that your manuscript has been deemed suitable for publication in PLOS ONE. Congratulations! Your manuscript is now with our production department. 

Kind regards, 

on behalf of

Professor Pilwon Kim 

Academic Editor

PLOS ONE